# The Roles of Hypoxia Imaging Using ^18^F-Fluoromisonidazole Positron Emission Tomography in Glioma Treatment

**DOI:** 10.3390/jcm8081088

**Published:** 2019-07-24

**Authors:** Kenji Hirata, Shigeru Yamaguchi, Tohru Shiga, Yuji Kuge, Nagara Tamaki

**Affiliations:** 1Department of Nuclear Medicine, Graduate School of Medicine, Hokkaido University, Sapporo 060-8638, Japan; 2Department of Neurosurgery, Graduate School of Medicine, Hokkaido University, Sapporo 060-8638, Japan; 3Central Institute of Isotope Sciences, Hokkaido University, Sapporo 060-0815, Japan; 4Department of Radiology, Graduate School of Medical Science, Kyoto Prefectural University of Medicine, Kyoto 602-8566, Japan

**Keywords:** hypoxia, imaging, glioma, fluoromisonidazole, FMISO, positron emission tomography, PET

## Abstract

Glioma is the most common malignant brain tumor. Hypoxia is closely related to the malignancy of gliomas, and positron emission tomography (PET) can noninvasively visualize the degree and the expansion of hypoxia. Currently, ^18^F-fluoromisonidazole (FMISO) is the most common radiotracer for hypoxia imaging. The clinical usefulness of FMISO PET has been established; it can distinguish glioblastomas from lower-grade gliomas and can predict the microenvironment of a tumor, including necrosis, vascularization, and permeability. FMISO PET provides prognostic information, including survival and treatment response information. Because hypoxia decreases a tumor’s sensitivity to radiation therapy, dose escalation to an FMISO-positive volume is an attractive strategy. Although this idea is not new, an insufficient amount of evidence has been obtained regarding this concept. New tracers for hypoxia imaging such as ^18^F-DiFA are being tested. In the future, hypoxia imaging will play an important role in glioma management.

## 1. Introduction

Several years have passed since we wrote two review articles regarding the role of hypoxia imaging for brain tumors [1] and head and neck cancers [2]. Herein, we again focus on the usefulness of hypoxia imaging in the treatment of brain tumors in order to introduce the more recently obtained findings. We also discuss the potential future uses of hypoxia imaging in cancer therapy.

## 2. Gliomas

Gliomas account for approx. 30% of all brain tumors and approx. 80% of all malignant brain tumors [3]. Patients with a glioblastoma, which is the most aggressive astrocytic tumor among the gliomas, have an especially poor prognosis with a 1-year survival rate of 56% despite the recent advances in surgery, radiotherapy, and chemotherapy treatments [4]. In glioma management, the precise diagnosis before the initial surgical resection followed by chemoradiotherapy is of great importance because the optimal treatment strategies and prognoses differ widely with the histological grade and expansion of the lesion. As gliomas often present with various grades of malignancy within the tumor, the detection of the most malignant part as well as the delineation of the entire tumor are essential in the preoperative evaluation.

In vivo imaging thus plays an important role in the treatment planning for a glioma. Genetic information has also become important. Magnetic resonance imaging (MRI) is the de facto standard modality for the initial evaluation of gliomas. MRI basically provides morphological or structural information. In contrast, positron emission tomography (PET) provides functional information by visualizing the metabolism of the cell, using radiotracers such as ^18^F-fluorodeoxyglucose (FDG) for glucose metabolism, ^18^F-fluorothymidine (FLT) for DNA replication, and ^11^C-methionine or ^18^F-fluoroethyltyrosine for amino acid metabolism. The grade of malignancy seems to reflect cell metabolism, and thus such metabolic imaging is a good choice for the detection of malignant parts of a lesion. PET can also visualize hypoxia. We next focus on the clinical significance of hypoxia and hypoxia imaging in glioma management.

## 3. Why Hypoxia Is Important Clinically

Hypoxia is defined as the insufficient concentration of oxygen. Hypoxia is a common feature of many different malignant tumors [5]. The microcirculation in tumors is often impaired by structural and/or functional disturbances of the tumor circulation, and rapid tumor growth may result in an increase in oxygen consumption, generating hypoxic conditions in which the oxygen supply is smaller than oxygen demand. Hypoxia is thus a phenomenon that has interested oncologists for many years.

In radiation therapy for cancer, the tumor oxygen concentration is an important factor that greatly affects the therapeutic effects. In theory, ionizing radiation interacts mainly with water molecules and less frequently with DNA molecules, which produce free radicals [6,7]. Under mediation by oxygen molecules, the free radicals fatally damage DNA molecules. The radiation sensitivity of cells is thus reduced under hypoxic conditions below a certain oxygen-concentration threshold.

One of the key players in hypoxic conditions is hypoxia-inducible factor 1 (HIF-1) [8]. HIF-1 is a transcription factor that helps cells survive under severely hypoxic conditions. Once hypoxia activates HIF-1, the molecule up-regulates a variety of genes controlling metabolic pathways, pH regulation, angiogenesis, metastatic potential, DNA replication, protein synthesis, and treatment resistance [6,7,9,10,11].

The expressions of HIF-1 and downstream molecules are negative prognostic factors in various malignancies [12]. The difference in radiosensitivity between hypoxic and normoxic cells has been called the oxygen enhancement ratio (OER). When using X-rays for radiotherapy, the OER is usually 2–3, which means that a two-to three-fold amount of radiation is needed in hypoxic conditions to obtain effects that are equivalent to those obtained in normoxic conditions. In addition to causing radioresistance, tumor hypoxia also interferes with the cytotoxic activities of many types of chemotherapy, a phenomenon known as chemoresistance [13].

Cancer cells have subpopulations with characteristics that are similar to those of nonmalignant stem cells. These cells are called ‘cancer stem cells’ [14,15]. HIF up-regulates Oct4 and c-Myc (both are Yamanaka factors that are important genes for stem cells). Since cancer stem cells are important targets for therapy, hypoxia imaging may help localize the cancer stem cells [16,17].

## 4. How the Oxygen Concentration Is Measured

There are a number of approaches to measure the tissue oxygen concentration (pO_2_) in vivo. Eppendorf polarographic oxygen microelectrode measurements have long been applied for this purpose in various cancers [18,19,20,21], but this technique is now rarely used in clinical settings because (1) it is invasive, (2) it is difficult to reach deep tissues with the needle used, (3) it demands a well-trained operator, and (4) the measurement itself may modify the oxygen concentration [22].

Another technique for measuring the oxygen concentration is immunohistochemical measurement using the bioreductive hypoxia marker pimonidazole [23]. Of course, this technique does not work in vivo, and it requires the tissue of interest to be sampled in some way. However, measurement using pimonidazole has been regarded as a gold standard for assessments of tumor hypoxia ex vivo. Noninvasive assessments of tumor hypoxia using various imaging methods have also been investigated. Blood oxygen level-dependent (BOLD) MR imaging can measure oxygen levels in blood using deoxyhemoglobin as an endogenous marker [23,24].

Positron emission tomography (PET) images reveal the distribution of positron-emitting tracers in the body. PET with hypoxia-imaging radiopharmaceuticals can be used clinically. Since PET has very high sensitivity and specificity compared to MR imaging and other imaging modalities, it enables the identification of regional hypoxia in vivo with an intravenous administration of a small amount of a radiopharmaceutical, in preclinical and clinical settings [9]. Although PET has spatial resolution that is inferior to that of immunohistochemical imaging, it permits the noninvasive delineation of hypoxic tissue in vivo.

## 5. What Is FMISO?

As mentioned above, PET is expected to become an ideal noninvasive tool for visualizing hypoxic conditions in vivo. The development of PET tracers for hypoxia has been conducted for decades. The first radiotracer was ^14^C-misonidazole as a beta-emitting tracer in 1981 [25], followed by ^18^F-fluoromisonidazole (FMISO) as a positron-emitting tracer [26]. To date, FMISO remains the most widely used radiotracer for imaging hypoxia in patients with brain tumors, although other radiotracers are being investigated [27].

The use of FMISO for brain tumors has a relatively long history. It was nearly 30 years ago that Valk et al. reported the first use of FMISO for glioma imaging, in 1992 [28]. However, FMISO was not used very frequently during the 1990s. After the dramatic increase in the number of PET-CT scanners in the first decade of the 21st century (due mainly due to the clinical usefulness of FDG), researchers and clinicians were encouraged to use FMISO to visualize the hypoxic status of gliomas as well as many other tumors.

Before focusing on the oncology uses of FMISO, we will take a brief look at its non-oncology uses. FMISO has been applied for ischemic cardiac and brain diseases. Ischemia is closely associated with hypoxia. As a cardiology application, Martin et al. used an ischemic canine myocardium model to determine the ability of FMISO to noninvasively visualize hypoxic but viable myocardium [29]. Their findings demonstrated that the myocardium with FMISO uptake had improved contraction after reperfusion, as expected. Regarding neurology, Read et al. investigated 24 stroke patients by FMISO PET imaging, and they observed the patients for up to 51 h to evaluate the prognosis [30]. They demonstrated that most of the tissues showing FMISO uptake in an acute phase proceeded to infarction and that the stroke patients’ neurological severity scores were positively correlated with hypoxic tissue volumes. These data suggest the usefulness of FMISO for predicting prognoses.

More recently, Alawneh et al. examined three acute stroke patients by multimodal imaging [31]. FMISO trapping was present mainly in the penumbra but also in the ischemic core, raising the possibility that there may be viable cells even in the ischemic core. In a pilot study by Fryer et al., normobaric hyperoxia (NBO) treatment was tested in a rodent ischemic stroke model to determine whether this treatment reduces FMISO uptake [32]. NBO did not appear to substantially reduce the size of FMISO uptake volume nor affect the FMISO kinetic rate constants. Although the evidence of the usefulness of FMISO continues to accumulate, FMISO is used less frequently in ischemic disease compared to cancer.

## 6. The Mechanisms of FMISO Uptake

The mechanisms of FMISO accumulation in hypoxic tissues have been described [33]. Once intravenously injected, FMISO is distributed to the cells via the blood flow. In the cells, the FMISO molecules capture electrons in the mitochondrial electron transfer system. In normoxic cells (i.e., cells without hypoxia), the FMISO electron is removed by oxygen molecules as a strong oxidant. FMISO that has lost the electron is excreted from the cells. In hypoxic cells (in which there is a lack of oxygen molecules) however, FMISO keeps extra electrons and stays in the cells. Therefore, FMISO is cleared from normoxic cells but accumulated in hypoxic cells.

Once accumulated, FMISO is irreversibly bound to high-weight molecules. FMISO is excreted from necrotic cells because there are no functioning mitochondria as an electron source in such cells. Masaki et al. reported another mechanism of FMISO accumulation [34]; they investigated the chemical forms of FMISO by using imaging mass spectrometry. A significant amount of the radioactivity in the examined tumor existed as low-molecular-weight compounds. The rate of clearance for each mode remains unknown. Masaki et al. also showed that the gluthathione conjugate of amino-FMISO is involved in FMISO accumulation in hypoxic tumor tissues, in addition to conventional mechanisms (i.e., binding to high-weight molecules). They also demonstrated that FMISO accumulation is dependent on glutathione conditions as well as the hypoxic status [35] (Figure 1).

The threshold of oxygen partial pressure that determines whether FMISO is accumulated or excreted is generally believed to be approximately 10 mmHg [36,37]. A 2016 clinical study by Veenith et al. evaluated 10 patients with traumatic brain injury with the use of ^15^O-labeled gas and FMISO; the threshold of FMISO accumulation was estimated as 15 mmHg [38]. It is important to keep in mind that FMISO accumulates only in such severely hypoxic tissues.

An advantage of FMISO to image brain, FMISO can go through the blood–brain barrier (BBB) because of its lipophilic nature. FMISO slightly accumulates in the normal brain tissues. In contrast, hydrophilic compounds, such as ^18^F-fluoroazomycin arabinoside (FAZA) and 1-(2,2-dihydroxymethyl-3-[^18^F]fluoropropyl)-2-nitroimidazole (DiFA) as mentioned later in this article, do not go beyond the BBB and do not accumulate in the normal brain [39]. Both lipophilic and hydrophilic agents can accumulate in a tumor where BBB is impaired (i.e., gadolinium-enhancing tumor), whereas only lipophilic agents can accumulate in a BBB-preserved tumor.

Since it is lipophilic, FMISO is excreted via the hepatobiliary system and thus physiologically accumulates in the liver and GI tract. Thus, FMISO is not efficient to evaluate hypoxic conditions of tumors in such organs. In contrast, the lung and the head and neck regions as well as the brain are the areas which do not show physiological FMISO uptake. Therefore, relatively large numbers of investigations regarding such tumors have been reported to show the clinical usefulness of FMISO. Especially for brain tumors, biopsy is so invasive that hypoxia imaging as the non-invasive method of tumor grading has a great clinical value.

## 7. The Clinical Usefulness of FMISO

### 7.1. Differential Diagnosis

Figure 2, Figure 3 and Figure 4 show some representative cases of low to high grade gliomas. We demonstrated that FMISO has the potential to distinguish glioblastomas (i.e., grade IV gliomas) from less-malignant gliomas (i.e., grade III or lower grade gliomas) [40]. As mentioned above, the oxygen concentration threshold of FMISO uptake is low. Thus, FMISO PET can differentiate tissues with severe hypoxia from those without. Studies using direct needle electrodes suggested that the hypoxic condition of a glioma depends on its degree of malignancy [41,42,43]. We used FMISO PET and FDG PET for the preoperative examination of 23 patients with gliomas of different World Health Organization (WHO) grades [40]. The PET findings were compared with the patients’ post-operative histological findings by neuropathologists. We observed FMISO uptake in the glioblastomas, but not in the less-malignant gliomas (Table 1). In the WHO definition, glioblastoma presents with necrosis in the tumor, whereas lower-grade gliomas do not develop necrosis [44]. It is thus reasonable that only glioblastomas have severe hypoxia (beyond the FMISO threshold) and therefore take up FMISO. We concluded that FMISO PET may be able to clearly distinguish glioblastomas from lower-grade gliomas.

There are several reports of the clinical value of FMISO PET in terms of the grading of gliomas. Cher et al. provided the FMISO PET findings of patients with gliomas of various grades [45], and they found that although all grade IV tumors showed a high FMISO uptake (which is consistent with our observation), only one of the three grade III gliomas showed FMISO uptake. Yamamoto et al. also observed FMISO uptake in some grade III gliomas, and they reported that the uptake in grade IV gliomas was significantly higher than in grade III or lower gliomas [46]. More recently, Kanoto et al. used FMISO PET to investigate 41 patients with WHO grade II–IV gliomas [47], and their findings indicated that a tumor-to-normal tissue ratio of 1.25 as a cut-off distinguished the patients’ grade II gliomas from the grade III and IV gliomas, with 90% sensitivity and 91% specificity.

Our study [40] and these previous investigations [45,46,47] are thus basically consistent but slightly different in that FMISO uptake was observed in grade III gliomas in the latter studies but not in ours. One of the reasons for this may be due to the uptake time of FMISO. After the intravenous FMISO injection in the other studies, the images were usually acquired at 2 h, whereas we acquired the PET images 4 h after injection. Thorwarth et al. discussed a theoretical problem with the 2-h imaging of FMISO [48]: using a kinetic analysis of a dynamic FMISO PET dataset, they reported that some of the hot spots on the 2-h FMISO images had disappeared on the 4-h FMISO images. This suggested that the high uptake on the 2-h images may have reflected a high initial influx of the tracer due to increased blood flow as well as hypoxia. In other words, the 4-h images can be expected to represent hypoxia alone, whereas the 2-h images would represent increased blood flow regardless of hypoxia. Two-hour imaging has advantages: (1) less time is required for the entire examination, reducing the patients’ burden, and (2) approximately twice the number of photons is theoretically detected by the PET scanner compared to 4-h imaging, considering that the half-life of ^18^F is 110 min. To further optimize the procedure, direct comparisons of these protocols in the same patient groups are needed. To further optimize the procedure, we directly compared 2-h vs. 4-h imaging in terms of the diagnostic performance of glioblastoma [49]. In a visual assessment of six glioblastoma lesions of six patients, four (67%) lesions were detected as positive uptake at 2 h, while six (100%) lesions were detected at 4 h. In our additional investigation (unpublished data), where we compared 2-h vs. 4-h imaging for five non-glioblastoma patients, all (5/5) showed slight uptake at 2 h but not at 4 h. However, a larger study is needed to reach a conclusion.

### 7.2. FMISO versus Necrosis, Vascularization, and Permeability

Several research groups have compared FMISO uptake with histological findings (such as necrosis and vascularization) and functional features (such as permeability). Our group investigated 59 patients with brain tumors including astrocytic tumors and metastatic tumors, and we examined the presence of microscopic necrosis [50]. In the visual analysis, 26 of the 27 FMISO-positive patients presented with necrosis, whereas 28 of the 32 FMISO-negative patients showed no necrosis. Using a cut-off of the tumor-to-normal tissue ratio, the presence of necrosis can be predicted with 96.7% sensitivity and 93.1% specificity. Our findings demonstrated that FMISO uptake is strongly associated with necrosis. It should be remembered that FMISO accumulates in viable tissues but not in the necrotic core. In case of tumors with vastly necrotic tissues, FMISO will not accumulate in the necrotic tissue itself, but will accumulate in the peri-necrotic tissues.

Watabe et al. used FMISO and ^15^O-labeled gas PET to investigate a rat model implanted with C6 glioma [51]. They observed correlations between the FMISO uptake and (1) the blood flow, (2) the metabolic rate of oxygen, and (3) the blood volume. The oxygen extraction fraction showed a significant increase in the severely hypoxic region compared to non-hypoxic and mildly hypoxic regions. Those authors concluded that intratumoral hypoxic regions showed decreased blood flow with increased oxygen extraction.

Ponte et al. studied glioblastoma patients by quantifying and localizing hypoxia with the use of FMISO PET, vascularization (cerebral blood volume [CBV]) by using MRI, and vascular permeability (contrast enhancement after gadolinium injection) by MRI [52]. Their analyses revealed that hypoxia and hypervascularization were correlated regarding both their maximum values and their volumes. A large proportion of the high CBVs co-located with hypoxia (>80% of the patients) and with contrast enhancement (>40% of the patients). Ponte et al. suggested that a strong association may exist between hypoxia and angiogenesis. There might be insufficient tumor oxygenation in human glioblastomas, despite increased tumor vascularization.

As with other studies, these lines of evidence help us understand the microenvironment of FMISO uptake in tissues.

### 7.3. The Prognostic Value of FMISO PET

As noted above, FMISO PET reflects the tumor microenvironment and is useful for the differential diagnosis of glioma. It is thus reasonable that an evaluation using FMISO could help predict the prognoses of glioma patients. In fact, several studies have addressed this question. For example, Lawrence et al. investigated 17 patients with suspected glioma and reported that positive FMISO uptake was associated with poor survival [45]. Spence et al. analyzed the results of FMISO PET performed for 22 glioblastoma patients before radiotherapy and compared the findings with the patients’ time to progression [53]. The results of their analyses demonstrated that greater hypoxia volume and a greater tumor-to-blood uptake ratio were associated with earlier progression. In a subsequent paper, Spence et al. also evaluated the predictive performance of several image features derived from FMISO PET and MR imaging [54]. They found that the most significant predictors of survival were the hypoxia volume, the hypoxia surface area, and the tumor-to-blood uptake ratio measured on FMISO PET images.

In a 2016 multicenter clinical trial reported by Gerstner et al., 42 patients with newly diagnosed glioblastoma were evaluated [55]. Tumor vasculature and hypoxia were visualized using MRI and FMISO PET, respectively. The survival analysis showed that a greater uptake of FMISO was associated with shorter overall survival. Other negative prognostic markers were increased tumor perfusion, vascular volume, and vascular permeability measured with MRI.

More recently, our group also evaluated the clinical value of FMISO PET for predicting the prognoses of glioma patients; a total of 32 glioblastoma patients underwent both FMISO PET and FDG PET [56]. Glucose metabolism can be visualized using FDG. In that study, we tested the hypothesis that the accumulation of both FMISO and FDG may express hypoxia and elevated glycolysis, i.e., anaerobic glycolysis. We applied a voxel-by-voxel analysis to the glioblastoma patients’ FMISO and FDG PET images. FDG positivity was defined as an FDG tumor-to-normal ratio (TNR) ≥1.0, and FMISO positivity was defined as an FMISO TNR ≥1.3. We introduced a new image-derived biological feature, the ‘metabolic tumor volume in hypoxia (hMTV)’ as the volume in which both FMISO and FDG are positive. The results of the survival analysis demonstrated that the hMTV was a predictive factor for both overall survival (OS) and progression-free survival (PFS), whereas the hypoxia volume (i.e., the FMISO-positive volume) alone was not a significant factor.

### 7.4. Treatment Response/Treatment Monitoring

Another role that is expected for FMISO PET is therapy monitoring. Since hypoxia is associated with tumor aggressiveness and poor prognosis, it is possible that changes in the FMISO uptake between pre- and post-treatment can be used to monitor a treatment’s effectiveness.

Our investigation of two glioblastoma patients revealed a considerable decrease in FMISO accumulation in the glioblastomas after chemoradiotherapy [57]. We speculated that reduced FMISO uptake may have reflected an increased oxygen concentration due to the radiotherapy, the so-called reoxygenation phenomenon.

Bevacizumab is a recombinant humanized monoclonal antibody that blocks angiogenesis by inhibiting vascular endothelial growth factor A (VEGF-A) [58]. Although bevacizumab failed to prolong the overall survival of patients with primary glioblastoma [59,60], it was effective in some populations of recurrent glioma patients. In our retrospective study, we investigated whether FMISO PET has the potential to distinguish responders to bevacizumab from non-responders [61]. Eighteen patients with recurrent glioma underwent bevacizumab treatment. We compared the patients’ pre-and post-MRI and FMISO PET to classify them as (1) MRI-FMISO double responders (*n* = 9, Figure 5), (2) MRI-only responders (*n* = 5, Figure 6), and (3) non-responders (*n* = 4, Figure 7). There were no FMISO-only responders. The survival analysis demonstrated that the MRI-FMISO double responders had significantly longer overall survival than the other patients, whereas no significant difference was observed between the MRI-only responders and the non-responders. We thus concluded that recurrent gliomas with decreasing FMISO accumulation after short-term bevacizumab application could derive a survival benefit from the treatment.

Wang et al. used C6 glioma model rats to evaluate the radiosensitization effects of oleanolic acid with FMISO [62]. They observed slowed tumor growth by the OA treatment combined with radiotherapy, as well as a decrease in FMISO uptake, and they suggested that FMISO PET can be of value for radiosensitization response evaluations.

### 7.5. Modifying Radiation Therapy Based on FMISO

Because hypoxia makes tumor cells insensitive to radiation therapy, an attractive strategy for hypoxic tumors is dose escalation. However, few studies have explored the clinical value of dose escalation in a multicenter prospective design; no such studies of brain tumors have been reported to our knowledge. A multicenter, prospective, randomized trial for lung cancer was conducted by Vera et al. [63]. In their study of 54 patients, they modified the radiation dose from the standard dose (66 Gy) used for the FMISO-negative patients to a high dose (up to 86 Gy) for FMISO-positive patients. They observed the dose escalation’s failure to improve the prognosis, but they later reported additional data in which the radiotherapy boost seemed to improve the FMISO-positive patients’ overall survival by 11.2 months [64].

Welz et al. reported the interim results of a similar trial for head and neck cancer [65]. The radiation dose was modified based on the patients’ FMISO PET findings, from the standard dose (70 Gy) to an escalated dose (77 Gy) targeting the hypoxic volume. They described the feasibility of dose escalation to the hypoxic volume without increasing toxicity. It seems that it will take some time to determine the clinical benefits and disadvantages of the application of a dose escalation based on hypoxia imaging in further multicenter prospective studies with larger numbers of patients.

## 8. The Reproducibility of FMISO PET Findings

Although FMISO has been used as a tracer for over two decades, the image reproducibility of FMISO PET has been demonstrated only recently. Why has the reproducibility been questioned? In addition to measurement errors, the degree of hypoxia can theoretically fluctuate. Hypoxia can generally be classified by the mechanism: (1) perfusion-related (acute) hypoxia due to insufficient blood flow, (2) diffusion-related (chronic) hypoxia caused by an increase in diffusion distances with tumor expansion, and (3) anemic hypoxia caused by a decrease in oxygen transport capacity [66]. Chronic hypoxia and anemic hypoxia are more stable than acute hypoxia, which can vary within a short time. In fact, a study using a model analysis and simulation showed that the use of FMISO PET could not distinguish acute hypoxia from chronic hypoxia [67]. Thus, the concern is that if hypoxia imaging reflects acute hypoxia rather than chronic hypoxia, the image reproducibility would not be guaranteed.

Nehmeh et al. addressed this question and reported a low reproducibility of FMISO uptake, both spatially and quantitatively [68]. They suggested that FMISO PET should not be used for radiation planning until further evidence has been gathered. However, one possible criticism of their study is that they calculated the hypoxic volume based on the tumor-to-blood ratio (TBR), and there has been no study evaluating the stability of radioactivity in the blood pool. Another problem regarding the study by Nehmeh et al. is that they acquired FMISO PET images after various lengths of time (i.e., 117–195 min), and such a variation has the potential to degrade the quantitative accuracy for any type of tracer.

In this context, we aimed to evaluate the reproducibility of FMISO uptake in head and neck cancers. We acquired high-contrast FMISO images for 11 patients on two occasions at a 48-h interval [69], and the results demonstrated that the FMISO uptake intensity (i.e., the TBR, TMR, and maximum standardized uptake value (SUVmax), the tumor volume, and the hypoxia location were quite reproducible. This reproducibility of tumor hypoxia is of clinical importance for optimum radiation planning and other treatment strategies.

Please note that hMTV (mentioned above in ‘The prognostic value of FMISO PET’ subsection), a volume showing both FDG and FMISO uptake, might reflect glucose metabolism upregulation as the result of hypoxia. It could be a biological response as an adaption to chronic hypoxia. In contrast, acute hypoxia might not allow cells to give enough time to increase glucose metabolism. That is our hypothesis to be demonstrated in future experiments.

## 9. The Quantification of Hypoxia Using FMISO

Thus far, we have focused on qualitative evaluations (visual assessment) and semi-quantitative (the SUVmax or tumor-to-normal ratio) evaluations by FMISO PET but as mentioned above, a tracer’s uptake can reflect both the tracer’s distribution (i.e., via the blood flow) and its specific binding (i.e., hypoxia). Quantitative approaches may have the potential to distinguish these two origins of uptake. Grkovski et al. used a rat xenograft model treated with cediranib or vehicle [70] and conducted a 90-min dynamic PET acquisition of FMISO before and after the treatment. A compartment model was applied to estimate the kinetic rate constants K1, k2, and k3, the distribution volume, and hypoxia-mediated entrapment. It was assumed that k3 reflected the irreversible binding of FMISO and thereby hypoxia. They observed that k3 was increased after the treatment, indicating increased hypoxia, but the uptake at 2 h was decreased, probably due to decreased blood flow as a treatment effect. This again indicates that tumors with higher vs. lower blood flow will show a different intensity of uptake at 2 h. We should pay attention when we interpret the 2-h images of FMISO PET.

These data suggest that a compartmental analysis may extract hypoxia information that enables an accurate evaluation of the hypoxic conditions despite an altered blood flow. Grkovski et al. applied the same compartmental analysis to head and neck cancer patients, and they reported that dynamic scanning using FMISO provided parametric maps of tumor hypoxia, perfusion, and radiotracer distribution volume, separately [71]. They also showed that the total acquisition times for obtaining clinically important information can be reduced to 20 min. Chakhoyan et al. later attempted to generate an absolute map of the tissue partial oxygen pressure in patients with glioblastomas [72]. Although such applications must be further validated in studies of larger populations, the high quantitative ability of PET along with biological information from radiotracers clearly has great potential.

## 10. The Current Hypoxia Imaging Challenges

We propose that FMISO has two major drawbacks. First, FMISO is not available in most of the world; it is commercially available in just a few countries (e.g., France) [63]. Many approaches using PET with a common tracer or other modalities such as MRI have been reported to predict regional hypoxia. Considering the wide availability of FDG, we examined the ability of FDG PET to predict FMISO uptake in head and neck cancer patients [73]. We hypothesized that hypoxia may increase the metabolic heterogeneity in the tumor and that a texture analysis conducted using FDG PET images may help extract hypoxia information. Even though we tested a wide variety of texture features, we unfortunately did not observe a satisfactory performance by the analysis. Valable et al. tested the performance of tissue oxygen concentration (S_t_O_2_)-MRI to visualize hypoxia [74]. They used several tumor model rats and investigated the performance of S_t_O_2_-MRI in a comparison with FMISO and pimonidazole staining, and they reported that S_t_O_2_-MRI was able to distinguish hypoxia from non-hypoxia with an area under the curve of 0.97.

Another drawback of using FMISO is that the clearance of FMISO from the blood is slow. This requires a long (~4 h) wait before scanning to acquire adequate hypoxia images. Although this is not a crucial disadvantage, hypoxia imaging would be more widely used than ever if the uptake time were shorter. As mentioned above, kinetic approaches would help shorten the protocol. Another approach is to modify the chemical structure of FMISO to make the clearance faster. Among such tracers, ^18^F-fluoroazomycin arabinoside (FAZA) seems to be the most extensively investigated, at least for gliomas [75,76,77]. The advantage of FAZA is its high hydrophilic nature, leading to its rapid clearance from the urinary system.

Similarly, we developed a new compound, 1-(2,2-dihydroxymethyl-3-[^18^F]fluoropropyl)-2-nitroimidazole (DiFA) that also has high hydrophilicity, and we investigated its uptake mechanisms [78]. We observed that DiFA was accumulated via a glutathione conjugation reaction in a manner that is similar to that of FMISO. We performed clinical trials using DiFA for the imaging of normal subjects and head and neck cancer patients [39].

## 11. Conclusions

We have focused in this review on glioma as the most common malignant brain tumor. Hypoxia is closely related to the malignancy of a tumor, and PET can noninvasively visualize the degree and the expansion of hypoxia. FMISO is currently the most commonly used radiotracer for hypoxia imaging. The clinical usefulness of FMISO PET has been established. FMISO PET can distinguish glioblastomas from lower-grade gliomas, and it can predict the microenvironment of tumors, including necrosis, vascularization, and permeability. FMISO PET provides prognostic information including survival and treatment response data. However, more evidence is needed before modifying radiation therapy based on FMISO PET. Next-generation tracers for hypoxia imaging, such as ^18^F-DiFA, are being tested. Hypoxia imaging may someday become a standard tool in glioma management.

## Figures and Tables

**Figure 1 jcm-08-01088-f001:**
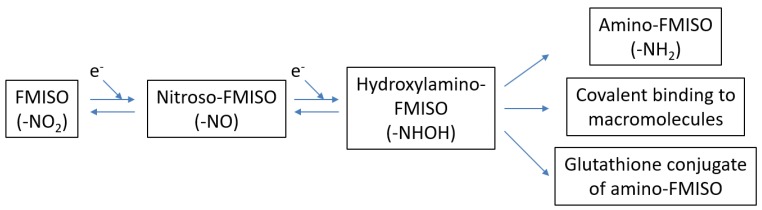
Proposed mechanisms [34] of the accumulation of ^18^F-fluoromisonidazole (FMISO). In hypoxic conditions, FMISO is reduced (i.e., FMISO obtains electrons) and either becomes amino-FMISO, covalently binds to macromolecules, or is conjugated with glutathione.

**Figure 2 jcm-08-01088-f002:**
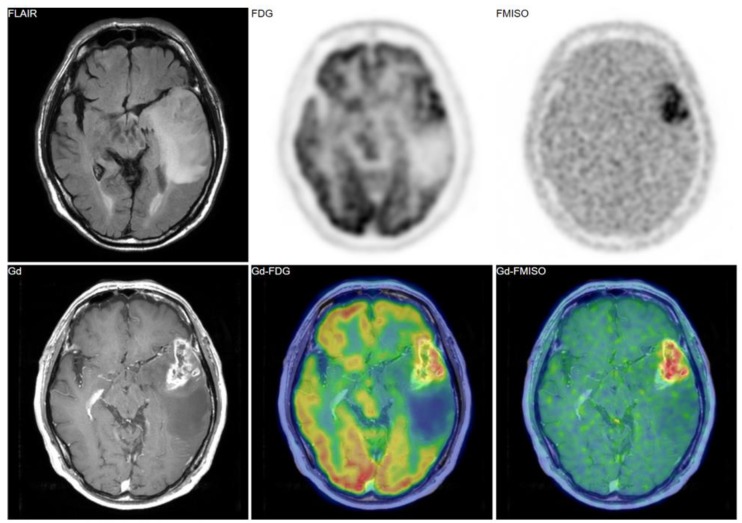
A 69-year-old patient had a tumor in the left temporal lobe. Fluid-attenuated inversion recovery (FLAIR) image showed high intensity, indicating the tumor and the peritumoral edema. Gadolinium enhancement, ^18^F-fluorodeoxyglucose (FDG) uptake, and FMISO uptake were observed in the same area. The pathological diagnosis was glioblastoma (grade IV).

**Figure 3 jcm-08-01088-f003:**
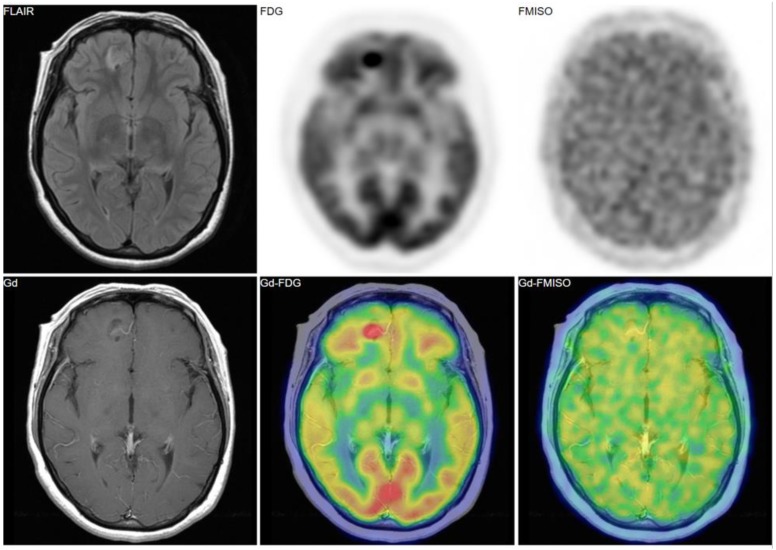
A 38-year-old patient had a tumor in the right frontal lobe. FMISO uptake was absent despite of strong uptake of FDG. The pathological diagnosis was gangliocytoma (grade I).

**Figure 4 jcm-08-01088-f004:**
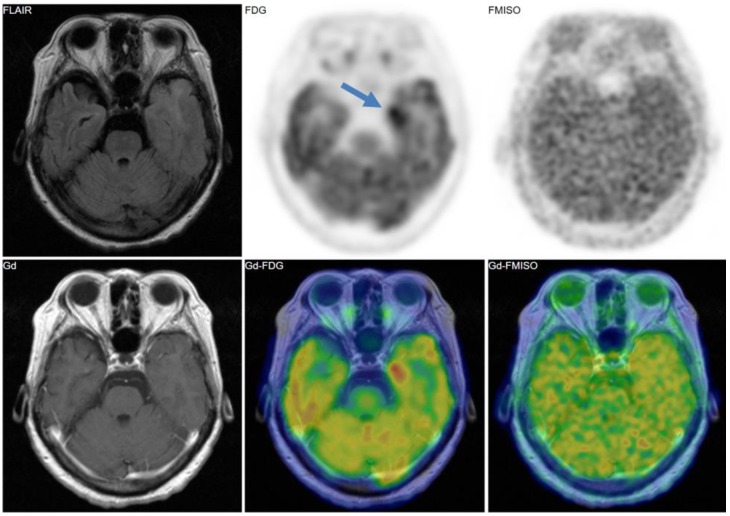
A 69-year-old patient had a tumor in the left medial temporal lobe. Like the case in Figure 3, FMISO uptake was not observed in the tumor, whereas FDG uptake was evident. The pathological diagnosis was anaplastic astrocytoma (grade III).

**Figure 5 jcm-08-01088-f005:**
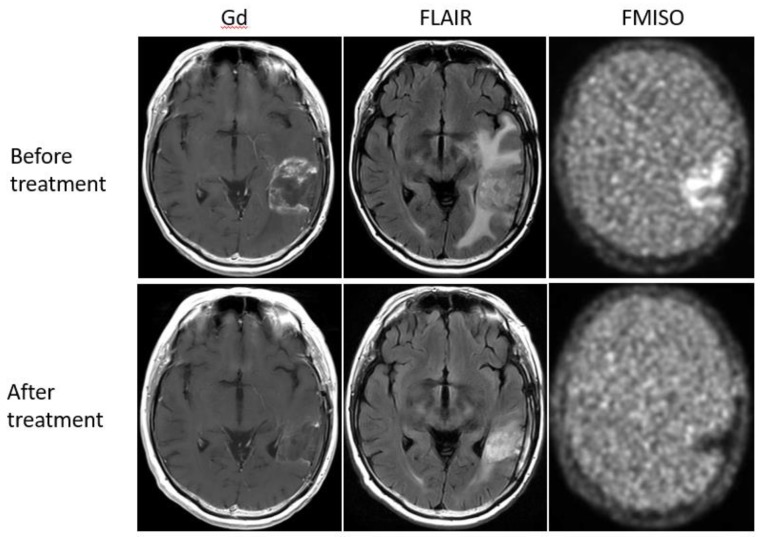
A 47-year-old patient with glioblastoma showed a strong uptake of FMISO before the bevacizumab treatment (upper row), but the FMISO uptake disappeared after the treatment (lower row). This patient was considered a ‘MRI-FMISO double responder’.

**Figure 6 jcm-08-01088-f006:**
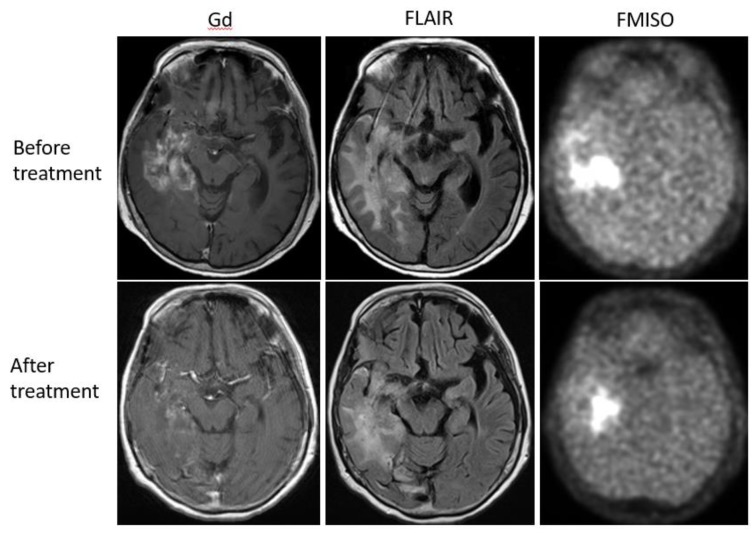
A 68-year-old patient with anaplastic astrocytoma showed a strong uptake of FMISO before the bevacizumab treatment (upper row). After the treatment, MRI showed shrinkage of the tumor, but FMISO uptake was still observed (lower row). The patient was considered a ‘MRI-only responder’.

**Figure 7 jcm-08-01088-f007:**
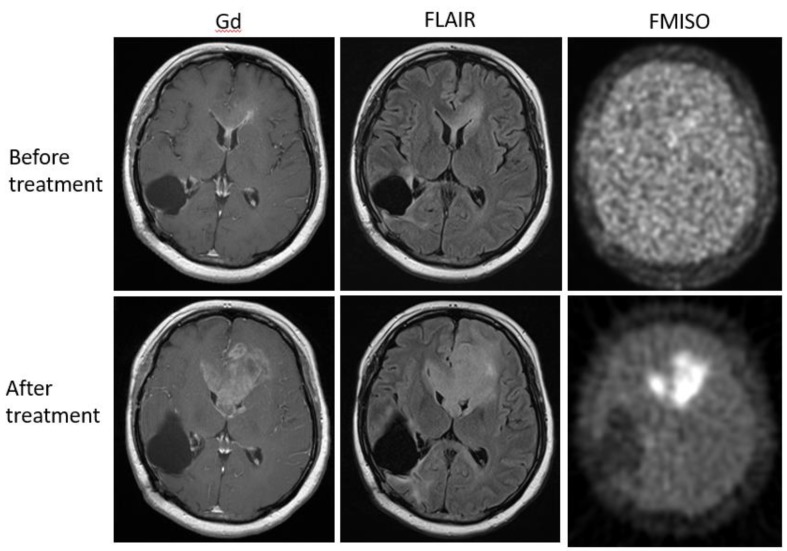
A 36-year-old patient with glioblastoma showed no FMISO uptake before the bevacizumab treatment (upper row). After the treatment, the patient presented with the enlarged tumor in the bilateral frontal lobe (lower row). The tumor showed high FMISO uptake. This patient was considered a ‘non-responder’.

**Table 1 jcm-08-01088-t001:** Correlation of FMISO uptake and pathological diagnosis.

	Post-Operative Pathological Diagnosis (Number of Patients)		
Visual assessment	Grade II	Grade III	Grade IV
FMISO uptake (−)	4	5	0
FMISO uptake (+)	0	0	14

*p* < 0.001 by Fisher’s exact test. The table was reconstructed from article [40].

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
