# Peer review of "The Roles of Hypoxia Imaging Using 18F-Fluoromisonidazole Positron Emission Tomography in Glioma Treatment"

_jcm, 2019, doi:10.3390/jcm8081088_

Reviewer 1 Report

Let me begin by saying what a pleasure it is to read a well written manuscript. The manuscript is valuable to the field and can be accepted in present form. However, it would be appreciated if the authors could address the following questions-

1) How long does FMISO remain bound to large biomolecules in the cell? Fig 1 shows the three modes by which it is retained in hypoxic tissue, is the rate of clearance of these three bound forms of FMISO known? At what point after delivery does semi-quantitative assessment by imaging lead to unreliable results. Page 9 desribes this to some extent but it seems the blood flow argument may not be valid in tumors with poor blood supply perhaps due to angiogenesis as described in the rat study reported in this review.

1a) What is the rate of false positives owing to the higher blood flow mediated accumulation at 2 hour timepoint?

2)While this study is focused on gliomas, a few statements on the difference FMISO uptake in gliomas vs other solid tumors would be helpful.

3)Page 9: Is it safe to assume that necrosis is a step beyond the grade 4 tumors? Is this is the case, FMISO should not accumulate in the necrotic tissue? So FMISO can detect microscopic necrosis but would give a false negative is the tissue were vastly necrotic?

4)Please comment on the ability of PET agents like FMISO to cross blood brain barrier.

5)In your opin ion would the hMTV method address the issue of acute vs chronic hypoxia as described in Nehmeh et al?

Author Response

Authors' Response to the Reviewers' Comments

We are very grateful to the editor and the two reviewers for the constructive comments, which we believe have substantially strengthened our manuscript. In the following pages, the reviewers' comments are underlined and italicized, and our responses are presented in plain text. In the revised manuscript, the modified passages are indicated with ‘Track Changes’ function.

Responses to the comments from Reviewer #1

Let me begin by saying what a pleasure it is to read a well written manuscript. The manuscript is valuable to the field and can be accepted in present form. However, it would be appreciated if the authors could address the following questions-

1) How long does FMISO remain bound to large biomolecules in the cell? Fig 1 shows the three modes by which it is retained in hypoxic tissue, is the rate of clearance of these three bound forms of FMISO known? At what point after delivery does semi-quantitative assessment by imaging lead to unreliable results. Page 9 desribes this to some extent but it seems the blood flow argument may not be valid in tumors with poor blood supply perhaps due to angiogenesis as described in the rat study reported in this review.

Authors response:

We thank the reviewer for raising this important issue. Unfortunately, we do not find any precise data regarding the rate of clearance of three biding forms of FMISO. Masaki et al. from our group suggested potential mechanisms of FMISO retention using imaging mass spectroscopy [1]. Accordingly, we added the following sentence in Page 4:

The rate of clearance for each mode remains unknown.

The optimum scanning timing for semi-quantitative assessment is discussed in “Differential diagnosis” subsection. To strengthen the discussion, we added the following sentences in “Differential diagnosis” subsection in Page 10:

To further optimize the procedure, we directly compared 2-hr vs. 4-hr imaging in terms of diagnostic performance of glioblastoma [49]. In visual assessment of 6 glioblastoma lesions of 6 patients, 4 (67%) lesions were detected as positive uptake at 2 hr, while 6 (100%) lesions were detected at 4 hr. In our additional investigation (unpublished data), where we compared 2-hr vs. 4-hr imaging for 5 non-glioblastoma patients, all (5/5) showed slight uptake at 2 hr but not at 4hr. However, a larger study is needed to reach conclusion.

We also thank the reviewer for the comment about tumors with poor blood flow. We discussed the related topic in “The quantification of hypoxia using FMISO” in Page 13. By an animal experiment, Grkovski et al. found that FMISO uptake at 2hr was reduced after chemotherapy, probably due to decreased blood flow. This indicates that tumors with poor blood flow may lower uptake than those with rich blood flow when compared at 2hr. Thus, we again think that we should pay attention when we interpret 2hr images of FMISO PET. Accordingly, we added the following sentences in Page 13:

This again indicates that tumors with higher vs. lower blood flow will show different intensity of uptake at 2 hr. We should pay attention when we interpret 2 hr images of FMISO PET.

1a) What is the rate of false positives owing to the higher blood flow mediated accumulation at 2 hour timepoint?

Authors response:

We thank the reviewer for asking the question. Previously, we directly compared 2-hr vs. 4-hr imaging of FMISO PET in terms of diagnostic performance of glioblastoma [2]. In visual assessment of 6 glioblastoma lesions of 6 patients, 4 (67%) lesions were detected as positive uptake at 2 hr, while 6 (100%) lesions were detected at 4 hr. In our additional investigation (unpublished data), we investigated 5 non-glioblastoma patients (i.e., they are not supposed to show FMISO uptake). All of them showed slight uptake at 2 hr but showed no uptake at 4hr. However, a larger study is needed to reach conclusion.

                 Accordingly, we added the following sentences in “Differential diagnosis” subsection in Page 11:

To further optimize the procedure, we directly compared 2-hr vs. 4-hr imaging in terms of diagnostic performance of glioblastoma [49]. In visual assessment of 6 glioblastoma lesions of 6 patients, 4 (67%) lesions were detected as positive uptake at 2 hr, while 6 (100%) lesions were detected at 4 hr. In our additional investigation (unpublished data), where we compared 2-hr vs. 4-hr imaging for 5 non-glioblastoma patients, all (5/5) showed slight uptake at 2 hr but not at 4hr. However, a larger study is needed to reach conclusion.

2)While this study is focused on gliomas, a few statements on the difference FMISO uptake in gliomas vs other solid tumors would be helpful.

Authors response:

We thank the reviewer for the valuable comment. FMISO physiologically accumulates in the liver and GI tract as the excretion system. Thus, FMISO is not efficient to evaluate hypoxic conditions of tumors in such organs. In contrast, the lung and the head-and-neck regions as well as the brain are the areas which do not show physiological FMISO uptake. Therefore, we can find relatively large numbers of investigations that reported clinical usefulness of FMISO, such as lesion characteristics and prognosis, for lung, head-and-neck, and brain tumors. Especially for brain tumors, biopsy is so invasive that hypoxia imaging as the non-invasive method of tumor grading has a great clinical value.

                 Accordingly, we added the following sentences in the following paragraph in Page 9:

Since lipophilic, FMISO is excreted via hepatobiliary system and thus physiologically accumulates in the liver and GI tract. Thus, FMISO is not efficient to evaluate hypoxic conditions of tumors in such organs. In contrast, the lung and the head-and-neck regions as well as the brain are the areas which do not show physiological FMISO uptake. Therefore, relatively large numbers of investigations regarding such tumors have been reported to show clinical usefulness of FMISO. Especially for brain tumors, biopsy is so invasive that hypoxia imaging as the non-invasive method of tumor grading has a great clinical value.

3)Page 9: Is it safe to assume that necrosis is a step beyond the grade 4 tumors? Is this is the case, FMISO should not accumulate in the necrotic tissue? So FMISO can detect microscopic necrosis but would give a false negative is the tissue were vastly necrotic?

Authors response:

We thank the reviewer for the valuable comment. As the reviewer mentioned, the existence of necrosis is essential in pathological diagnosis of glioblastoma (i.e., grade IV). Also, FMISO should not accumulate in the necrotic tissues or cells, but should accumulate in the peri-necrotic tissues showing hypoxia. In case of vastly necrotic tumor, FMISO should not accumulate in the necrotic core itself, but should accumulate in the viable tissues surrounding the necrotic tissues. As we reported [3], 28 of 32 (88%) FMISO-negative patients showed no necrosis.

                Accordingly, we added the following sentences in “FMISO versus necrosis, vascularization, and permeability” subsection in Page 11.

It should be remembered that FMISO accumulates in viable tissues but not in the necrotic core. In case of tumors with vastly necrotic tissues, FMISO will not accumulate in the necrotic tissue itself, but will accumulate in the peri-necrotic tissues.

4)Please comment on the ability of PET agents like FMISO to cross blood brain barrier.

Authors response:

We thank the reviewer for the valuable comment. FMISO is a lipophilic compound and thus it can go through the blood brain barrier (BBB). FMISO slightly accumulates in the normal brain tissues. In contrast, hydrophilic compounds, such as FAZA and DiFA [4], do not go beyond the BBB and do not accumulate in the normal brain. However, once BBB is impaired, such as in high-grade tumors, hydrophilic compounds can accumulate.

                 Accordingly, we added in the ‘The mechanisms of FMISO uptake’ subsection in Page 9:

An advantage of FMISO to image brain, FMISO can go through the blood brain barrier (BBB) because of its lipophilic nature. FMISO slightly accumulates in the normal brain tissues. In contrast, hydrophilic compounds, such as 18F-fluoroazomycin arabinoside (FAZA) and 1-(2,2-dihydroxymethyl-3-[18F]fluoropropyl)-2-nitroimidazole (DiFA) as mentioned later in this article, do not go beyond the BBB and do not accumulate in the normal brain [39]. Both lipophilic and hydrophilic agents can accumulate in a tumor where BBB is impaired (i.e., gadolinium-enhancing tumor), whereas only lipophilic agents can accumulate in a BBB-preserved tumor.

5)In your opinion would the hMTV method address the issue of acute vs chronic hypoxia as described in Nehmeh et al?

Authors response:

We thank the reviewer for the valuable comment. The idea is very interesting. hMTV (= metabolic tumor volume in hypoxia), showing both FDG and FMISO positive, might reflect glucose metabolism upregulation as the result of hypoxia. It could be a biological response and adaption. It could occur in a chronic hypoxia. In contrast, in acute hypoxia, there may not be enough time to adapt to hypoxia, and thus no glucose comsumption elevation. However, the story is only speculation and to be demonstrated in experiments.

                 Accordingly, we added the following paragraph in Page 16:

Please note that hMTV (mentioned above in ‘The prognostic value of FMISO PET’ subsection), a volume showing both FDG and FMISO uptake, might reflect glucose metabolism upregulation as the result of hypoxia. It could be a biological response as an adaption to chronic hypoxia. In contrast, acute hypoxia might not allow cells to give enough time to increase glucose metabolism. That is our hypothesis to be demonstrated in future experiments.

1.              Masaki Y, Shimizu Y, Yoshioka T, Tanaka Y, Nishijima K, Zhao S, Higashino K, Sakamoto S, Numata Y, Yamaguchi Y, Tamaki N, Kuge Y. The accumulation mechanism of the hypoxia imaging probe "FMISO" by imaging mass spectrometry: possible involvement of low-molecular metabolites. Sci Rep. 2015;5:16802.

2.              Kobayashi K, Hirata K, Yamaguchi S, Kobayashi H, Terasaka S, Manabe O, Shiga T, Magota K, Kuge Y, Tamaki N. FMISO PET at 4 hours showed a better lesion-to-background ratio uptake than 2 hours in brain tumors. Journal of Nuclear Medicine. 2015;56:373.

3.              Toyonaga T, Hirata K, Yamaguchi S, Hatanaka KC, Yuzawa S, Manabe O, Kobayashi K, Watanabe S, Shiga T, Terasaka S, Kobayashi H, Kuge Y, Tamaki N. (18)F-fluoromisonidazole positron emission tomography can predict pathological necrosis of brain tumors. Eur J Nucl Med Mol Imaging. 2016;43:1469-76.

4.              Watanabe S, Shiga T, Hirata K, Magota K, Okamoto S, Toyonaga T, Higashikawa K, Yasui H, Kobayashi J, Nishijima KI, Iseki K, Matsumoto H, Kuge Y, Tamaki N. Biodistribution and radiation dosimetry of the novel hypoxia PET probe [(18)F]DiFA and comparison with [(18)F]FMISO. EJNMMI research. 2019;9:60.

Reviewer 2 Report

This is a well written and thoroughly studied review article. The introduction and background are reasonable given the premise of the article. I would recommended the article for publications but with modification (minor revision). This review articles mainly focused on glioma as the most common malignant brain tumor. Authors made a comprehensive reports on clinical usefulness of 18F-fluoromisonidazole (FMISO) as a PET radiotracer in hypoxia imaging.

Though there are many PET radiotracer available for hypoxia imaging, this review article is specific with FMISCO PET. I would recommended authors to include FMISCO PET in title of this review article.

Author Response

Authors' Response to the Reviewers' Comments

We are very grateful to the editor and the two reviewers for the constructive comments, which we believe have substantially strengthened our manuscript. In the following pages, the reviewers' comments are underlined and italicized, and our responses are presented in plain text. In the revised manuscript, the modified passages are indicated with ‘Track Changes’ function.

Comments from Reviewer 2

This is a well written and thoroughly studied review article. The introduction and background are reasonable given the premise of the article. I would recommended the article for publications but with modification (minor revision). This review articles mainly focused on glioma as the most common malignant brain tumor. Authors made a comprehensive reports on clinical usefulness of 18F-fluoromisonidazole (FMISO) as a PET radiotracer in hypoxia imaging.

Though there are many PET radiotracer available for hypoxia imaging, this review article is specific with FMISCO PET. I would recommended authors to include FMISCO PET in title of this review article.

Authors response:

We thank the reviewer for understanding our manuscript. As recommended, we modified the title of the manuscript as:

“The roles of hypoxia imaging using 18F-fluoromisonidazole positron emission tomography in glioma treatment.”